# Arrhythmia classification detection based on multiple electrocardiograms databases

**Meng Qi[1,2], Hongxiang Shao[1,2]***, **Nianfeng Shi[1,2], Guoqiang Wang[1,2], Yifei Lv[3]**

**1** Computer and Information Engineering Department, Luoyang Institute of Science and Technology, Luoyang, China, **2** Henan Province Engineering Research Center of Industrial Intelligent Vision, Luoyang, China, **3** School of Computer Science and Engineering Department, Tianjin University of Technology, Tianjin, China

* 200900501667@lit.edu.cn

**Data Availability Statement:** Publicly available datasets were analyzed in this study. This data can be found here: https://physionet.org/about/database/.

## Abstract

According to the World Health Organization, cardiovascular diseases are the leading cause of deaths globally. Electrocardiogram (ECG) is a non-invasive approach for detecting heart diseases and reducing the risk of heart disease-related death. However, there are limited numbers of ECG samples and imbalance distribution for existing ECG databases. It is difficult to train practical and efficient neural networks. Based on the analysis and research of many existing ECG databases, this paper conduct an in-depth study on three fine-labeled ECG databases, to extract heartbeats, unify the sampling frequency, and propose a self-processing method of heartbeats, and finally form a unified ECG arrhythmia classification database, noted as Hercules-3. It is separated into training sets (80%) and testing sets (the remaining 20%). In order to verify its capabilities, we have trained a 16-classification fully connected neural network based on Hercules-3 and it achieves an accuracy rate of up to 98.67%. Compared with other data processing, our proposed method improves classification recall by at least 6%, classification accuracy by at least 4%, and F1-score by at least 7%.

## 1 Introduction

According to the World Health Organization, nearly 25.6 million people died from cardiovascular disease in 2020 [1, 2]. Cardiovascular diseases account for the most important deaths globally, with heart disease and stroke accounting for 80% of these deaths [3]. In order to reduce the risk of death, Electrocardiogram (ECG) [4], Phonocardiogram (PCG) [5], Magnetic Resonance Angiography (MRA) [6], Echocardiography [7], X-ray angiography [8] and 3D Holography [9] are effective detection approaches for heart disease in the early stages. Arrhythmia classification detection based on ECG databases has been the focus of research in recent years. With the development and advancement of artificial intelligence technology, autonomous doctor machines [10] will emerge, and the discharge time can be predicted based on health record data [11], which will greatly reducing the workload of doctors.

An abnormal alteration in the regular heart is the hallmark of the cardiovascular, known as arrhythmia. Arrhythmia can be fatal or at least dangerous [1]. Traditional and machine-learning-based techniques [12–14] of arrhythmia classification have been proposed by

**Funding:** This research was funded by the National Natural Science Foundation of China under Grant (No. 62176113), as well as the science and technology breakthrough project of the Henan science and technology department (No. 222102210094). The funders played a role in data collection and analysis, comprehensively reviewing and checking the final published papers, and making the final version.

**Competing interests:** The authors have declared that no competing interests exist.

researchers in recent years. In terms of arrhythmia classification, it has been proved that machine-learning techniques such as deep learning (and their derivatives) outperform traditional techniques [15, 16].

A number of research institutions have established databases for heartbeat classification detection. For instance, the MIT-BIH Arrhythmia Database(MIT-BIH) [17], the MIT-BIH Supraventricular Arrhythmia Database (MIT-BIH-Sup) [18], and the St Petersburg INCART 12-lead Arrhythmia Database (St-Peterburg [19], and Creighton University Ventricular Tachycardia database are the most commonly used databases for evaluating the ECG signals. These databases have been used in many papers to evaluate the performance of classification neural networks. It is challenging to get a high-performance classification network due to the existing single database having limited samples and imbalanced sample distribution. Due to the differences between ECG databases, such as the number of leads, sampling frequency, and data encoding, it is not feasible to directly merge databases and use them to train and evaluate classification networks.

To address this problem, we conducted studies of several popular ECG databases, including MIT-BIH, MIT-BIH-Sup, and St-Petersburg, based on analyzing many existing ECG databases. As part of our research, we extracted heartbeats, resampled them under the same frequencies, and presented a method for self-preprocessing heartbeats, described as Hercules-3, to create a unified ECG heartbeat classification database. We partitioned this database into training sets and test sets in a ratio of 8 to 2. To verify the effectiveness of this database, we designed a five-layer fully connected deep network and achieved a 98.63% accuracy rate of 16-class heartbeats.

The main innovative work and conclusions of this paper are as follows:

(1) We provide a strategy for integrating several ECG databases, extracting heartbeats from each of the three ECG databases and partitioning them proportionally into training and testing sets to form a unified database that can be used for heartbeat classification detection.

(2) Depending on the properties of ECG signals, a heartbeat self-processing method is proposed to eliminate the differences between heartbeats inter-databases and intra-database effectively. Our analysis revealed that our approach is more effective than traditional preprocessing methods.

(3) To verify the effectiveness of the database, we designed a 5-layers fully connected structure of deep network for classifying 16 different kinds of heartbeats and got an accuracy of up to 98.63%. This is, to the best of our knowledge, the greatest classification accuracy for 16-class heartbeats. To verify model performance, Sudden cardiac death Holter database is used for an inter-patient experiment. All heartbeats of this database were viewed as test data, and the accuracy of the fully connected network structure test was 96.62%. There are two reasons for such high accuracy. First, the integration of three databases makes the heartbeats number significantly increased, and the complementarity of heartbeats between the databases solves the problem of imbalance distribution for multiple classes to some extent. Second, the heartbeat self-processing method effectively eliminates sample interference factors while preserving sample features.

The basic architecture of this paper is as follows: Section II analyzes recent research in terms of heartbeats classification detections. Section III proposes an integration method of multiple ECG databases and describes its processes. Section IV designs a five-layer fully connected neural network for 16-class heartbeats detection. Section V analyzes the heartbeat classification results to verify the effectiveness of the integrated database. The conclusion is shown in section VI.

## 2 Related works

As noted in the preceding section, several machine-learning-based techniques of heartbeat classification have been presented in the literature.

In [15], Dinakarrao, S.M.P. et al. classified the arrhythmia detection approaches and discussed their performances, complexities, and suitability for hardware implementation. The parameters used for evaluating arrhythmia detection performance are accuracy, sensitivity, and specificity.

In [20], a classification approach for heartbeats utilizing a combination of morphological and kinetic characteristics was presented. The suggested technique has 99.3% accuracy in the "class-oriented" assessment and 86.4% accuracy in the "subject-oriented" evaluation on the MIT-BIH arrhythmia database.

In [21], they used one against rest method to classify 16 different arrhythmias, but its database actually has only ten kinds of heartbeats and it has reached an overall accuracy of 88.24%. In citeref11, three decision trees, random forest, and logistic regression classifiers were developed. The performance of the random forest classifier was much higher than that of the decision trees and logistic regression. It uses the MIT-BIH database to achieve 88% accuracy for 3-class classifications.

In [22], three decision trees, random forest, and logistic regression classifiers were developed. The performance of the random forest classifier was much higher than that of the decision trees and logistic regression. It uses the MIT-BIH database to achieve 88% accuracy for 3-class classifications.

In [23], a proprietary convolutional neural network, called ArrhyNet, for arrhythmia classification, was proposed and evaluated by the MIT-BIH arrhythmia database. Several data processing and augmentation methods, such as the low pass filter, baseline wander filter, and Synthetic Minority Over Sampling (SMOTE), are used to increase the accuracy rate. The top-1 accuracy for sixteen distinct kinds of arrhythmias is 92.73%.

In [24], Pooja Sharma et al. propose a Linear Adaptive Sine–Cosine Algorithm, which is an effective deep neural network method for enhancing the segmentation method of ECG signals for accurately classifying 16-class arrhythmia disease. The suggested SVM&DNN with LA-SCA achieves 99.29% accuracy, 97.51% sensitivity, and 98.66% specificity across complete data.

The machine-learning-based heartbeat classification techniques have not been adopted by healthcare professionals, mainly because of the imbalanced data for classification [1]. In addition to the imbalance of data, recording numbers of several heartbeat classes are too small to contain the main features of their classes, resulting in deep networks cannot comprehensively extract the features of their classes. To solve these problems, we integrated multiple complementary ECG databases which labeled the heartbeats classes into a unified database by extracting heartbeats under the same or nearby lead, and then unified sampling frequency by resampling techniques, and then self-processing to eliminate noise intra-databases and interdatabases. It greatly increases the number of several heartbeat classes, especially arrhythmias that originally had a small number of heartbeats.

## 3 ECG databases integration methodology

### 3.1 ECG databases

ECG databases are the most important data for training and evaluating neural networks for arrhythmia classification. The following ECG databases are often used for evaluating classification networks: MIT-BIH database, MIT-BIH-Sup database, St. Petersburg database, and so on.

In addition, some ECG databases are used for the evaluation of tachycardia or bradycardia detection methods, such as the Creighton University Ventricular Tachycardia database. Tachycardia and bradycardia are relatively independent special heart rhymes, which are not considered in this paper.

MIT-BIH database is the most widely used ECG database in heartbeat classification and arrhythmia detection research. There are 48 half-hour snippets of 2-channel ECG recordings made from 47 people that the BIH Atrial fibrillation Laboratory analyzed in the document [15]. Each of its heartbeats is marked with the R peak and the beat type. The original database has 16 heartbeat classifications. Five classifications are suggested by the Association for the Advancement of Medical Instrumentation for these various heartbeat patterns. Therefore, many researchers classify heartbeats into five categories in their papers. This greatly reduces the difficulty of heartbeat classification. According to our analysis, the more heartbeat types, the more difficult it is to classify. MIT-BIH is labeled with at least 16 heartbeat classes, as shown in Table 1. To avoid duplication, parent classes that contain other kinds are removed. For example, the bundle branch block beat (Noted as BBB) is a superclass containing the left bundle branch block beat (Noted as LBB), and the right bundle branch block beat (Noted as RBB) is removed.

MIT-BIH Supraventricular Arrhythmia Database (MIT-BIH-Sup) includes 78 half-hour ECG recordings chosen to supplement the examples of supraventricular arrhythmias in the MIT-BIH Arrhythmia Database. The St Petersburg INCART 12-lead Arrhythmia Database has 75 annotated 30-minute recordings with 12 standard leads sampled at 257 Hz. The heartbeats of the above three ECG databases were extracted, and the number of heartbeats of each class was counted. Table 1 displays the statistical results.

According to the statistical results, the MIT-BIH database contains 15 kinds of heartbeats with a total of 109494 heartbeats, of which 75052 normal beats (NM), 33 unclassifiable beats (UC), and only two supraventricular premature beats (SP). There are too few samples for some classes of heartbeats, such as SP and UC, so MIT-BIH cannot independently support the

**Table 1. Heartbeats classes of MIT-BIH arrhythmia databases.**

| Heartbeat classes | Abbr. | MIT-BIH | MIT-BIH-Sup | ST-Petersburg | Summary |
|---|---|---|---|---|---|
| Normal beat | NB | 75052 | 162339 | 150410 | 1993143 |
| Fusion of paced and normal beat | FPNB | 982 | | | 1394 |
| Atrial escape beat | AEB | 16 | | | 16 |
| Paced beat | PB | 7028 | | | 30151 |
| Nodal (junctional) escape beat | NEB | 229 | | 92 | 321 |
| Supraventricular escape beat | SEB | | | 32 | 37 |
| Left bundle branch block beat | LBBB | 8075 | | | 8075 |
| Right bundle branch block beat | RBBB | 7259 | | 3174 | 10433 |
| Supraventricular premature | SP | 2 | 12188 | 16 | 14498 |
| Atrial premature beat | APB | 2546 | | 1944 | 4490 |
| Nodal (junctional) premature beat | NPB | 83 | 9 | | 1601 |
| Aberrated atrial premature beat | AAPB | 150 | 1 | | 153 |
| Premature ventricular contraction | PVC | 7130 | 9943 | 20013 | 65475 |
| Ventricular escape beat | VEB | 106 | | | 122 |
| Fusion of ventricular and normal beat | FVNB | 803 | 23 | 219 | 1708 |
| Unclassifiable beat | UB | 33 | 79 | 6 | 211 |
| Summary | | 109494 | 184583 | 175907 | 2186555 |

training and testing of 15-class classification networks. The same goes for the MIT-BIH-Sup database and ST-Petersburg database. MIT-BIH-Sup contains 8-class heartbeats, of which the number of AAP heartbeats is only 1. ST-Petersburg contains 9-class heartbeats, of which the number of UC is only 6. Therefore, none of the three databases can support the training and testing of the 16-class heartbeats classification network. Therefore, most papers investigate the five categories of ECG heartbeat classifications according to the AAAI standard.

In addition, the analysis shows that there is a certain degree of complementarity between the above three databases. For example, the samples of SP heartbeats in MIT-BIH-Sup can effectively compensate for the shortage of SP heartbeats in the MIT-BIH database. SE heartbeats in St-Petersburg can effectively compensate for the empty data of SE heartbeats in the MIT-BIH database. Additionally, the tiny sample classes NE and UC in the MIT-BIH database have been augmented by combining MIT-BIH-Sup and ST-Petersburg. Therefore, integrating multiple databases together makes a lot of sense, making each class of heartbeats relatively abundant. In addition, the Sudden Cardiac Death Holter Database will be used to validate the performance of a neural network model trained on data from the integration of the three databases. It is a collection of long-term ECG recordings of patients who experienced sudden cardiac death during the recordings. These recordings were mainly obtained in Boston hospitals. The database currently includes 23 patients with underlying sinus rhythm, continuously paced, or atrial fibrillation.

## 3.2 Difference analysis between databases

The most direct way to integrate multiple databases is to merge the databases directly without any processing for ECG heartbeats. However, these databases contain ECG data collected by multiple devices at different sampling frequencies and different leads for different populations. There are obvious differences in terms of data characteristics among different databases. It is not feasible to simply merge multiple databases into a large database. Table 2 shows information such as sampling frequency and lead of each database.

Different sampling frequencies mean that the same number of sampling points represents different heartbeat durations. For a given number of single heartbeat samples, if the sampling frequency is too large, the heartbeat may not be fully extracted, resulting in partial heart waveform data loss. Otherwise, if the sampling frequency is too low, it may result in the extracted heartbeats containing part of the data of adjacent heartbeats. Neither of these conditions is conducive to training neural networks, which may lead to low efficiency in arrhythmia recognition and heartbeat classification. In addition, the data collected by different leads have certain correlations and obvious differences. To this end, we want to unify the lead and avoid data discrepancies due to different leads. Different devices and different acquisition objects will also lead to significant differences in the amplitude of the waveform.

Table 3 shows the mean, standard deviation, minimum, median, and maximum values of the two ECG databases and their subsets in detail. From this table, there are obvious differences in the statistical characteristics of different databases. For instance, The R-value for

**Table 2. Data information of ECG databases used in this work.**

| Database name | Sampling Frequencies | Leads |
|---|---|---|
| MIT-BIH Arrhythmia Database | 360 | MLII, V5 |
| MIT-BIH supraventricular Arrhythmia Database | 128 | 12-leads |
| ST Petersburg Incart 12-lead Arrhythmia Database | 257 | V4, MLIII |

**Table 3. Data statistics of different databases used in this work.**

| Data | Num. of Data | Mean | Std Dev | Min | Median | Max |
|---|---|---|---|---|---|---|
| MIT-BIH R-value | 109450 | 0.9823 | 0.8603 | -5.12 | 1.105 | 5.115 |
| ST-Petersburg R-value | 175813 | 0.9034 | 3.559 | -17.78 | 0.8088 | 18.26 |
| MIT-BIH NE R-value | 229 | 0.7887 | 0.2838 | -0.05 | 0.81 | 1.415 |
| ST-Petersburg NE R-value | 92 | -1.073 | 1.4608 | -3.94 | -1.4498 | 2.3695 |

St. Petersburg goes from -17.78 to 18.26, while the maximum value of the MIT-BIH NEB R-value is only 1.415 and a minimum value of -0.05. This means that the two databases cannot be merged directly together and must be preprocessed.

## 3.3 Data pre-process

In addition to factors such as collection objects and collection devices, interference is another important cause of differences between databases. Interference is also known as noise. There are three major noises in ECG signals, namely baseline wander, power frequency interference, and electromyography interference.

Baseline wander means that the measured signal will shake up and down due to the measurement electrodes and the human body's breathing, making it difficult to locate the feature points of the ECG signal and reducing data availability. Power frequency interference is a problem that almost all analog circuits will encounter. As long as the ECG equipment is exposed to the air without a metal shield, the power frequency interference problem cannot be avoided. If the signal strength is relatively large, such as above 1V, the power frequency interference may have little effect. If the signal is below 1mV, power frequency interference can affect long lead wires on ECG equipment. Due to the mixture of various bioelectricities in the human body, one bioelectricity is sometimes a useful signal, but it may be a useless noise in another situation. The human body's electricity other than the measured bioelectricity is noise. Myoelectric interference is caused by human muscle vibration, which is random, and the frequency range is between 5 and 2000 Hz.

To eliminate the differences between databases, we compare the multiple databases described above and employ multiple data pre-processing methods. Traditional data pre-processing main standardization, extremum, etc. To explain the data pre-processing method more intuitively, suppose that our ECG heartbeats data set is A, which contains m heartbeats in total, and each heartbeat is composed of n sampling points, then A can be expressed as the following matrix:

$$A = \begin{bmatrix} a_{0,0} & a_{0,1} & \cdots & a_{0,n-1} \\ a_{1,0} & a_{1,1} & \cdots & a_{1,n-1} \\ \vdots & \vdots & \ddots & \vdots \\ a_{m-1,0} & a_{m-1,1} & \cdots & a_{m-1,n-1} \end{bmatrix}$$

Traditional methods process data according to feature dimensions to eliminate quantitative differences between different feature dimensions. Common pre-processing methods include standardization, extreme value, averaging, and standard deviation. Database A is represented

by the feature dimension as:

$$A = \begin{bmatrix} B_0 & B_1 & \dots & B_{n-1} \end{bmatrix}$$

$$B_j = \begin{bmatrix} a_{0,j} & a_{1,j} & \dots & a_{i-1,j} \end{bmatrix} \quad 0 \le j \le n-1$$

(a) Standardization. Standardization is carried out in units of databases or data batches. Thus, the mean and standard deviation of the dataset or data batch are usually calculated first, and then the standard deviation is used to divide each sample by the difference from the mean. Following the use of standardization, the variable has zero mean and one standard deviation. The standardized formulation is as follows:

$$a'_{i,j} = \frac{a_{i,j} - \bar{a}_J}{s_j} \tag{1}$$

$$\bar{a}_J = \frac{1}{m} \sum_{t=0}^{m-1} a_{t,j} \tag{2}$$

$$s_j = \sqrt{\frac{1}{m-1} \sum_{t=0}^{m-1} a_{t,j} - \bar{a}_J} \tag{3}$$

As can be seen from Eqs (2) and (3), $\bar{a}_J$ and $s_j$ indicate the mean and standard deviation of the j+1th column of matrix A, respectively.

(b) Extreme value. The extreme value method usually transforms the original data into data within a specific range by taking the maximum and minimum values of the variables, thereby eliminating quantitative differences. The mathematical formulation of the extreme value method is as follows:

$$a'_{i,j} = \frac{a_{i,j} - min(a_j)}{max(a_j) - min(a_j)} \tag{4}$$

$$\mathbf{a_j} = \begin{bmatrix} a_{0,j} & a_{1,j} & \dots & a_{m-1,j} \end{bmatrix} \quad 0 \le j \le n-1$$

(c) Averaging. The mean method is to directly divide a variable by the mean of that variable. Unlike normalization methods, averaging methods can retain information about the degree of difference in values between variables.

(d) Standard deviation. It is a variation of the standardization method that directly divides the variable value by the standard deviation, rather than removing the mean and dividing by the standard deviation.

(e) our pre-processing method

The above methods all process data according to the feature dimension, which cannot eliminate the data difference between each heartbeat. Therefore, we propose a heartbeat self-processing method, that is, data processing according to the heartbeat dimension. The heartbeat self-processing method is to process the data in the unit of heartbeat, that is, the difference between heartbeat values and the average is divided by standard deviation. Its formulation is

as follows:

$$a'_{i,j} = \frac{a_{i,j} - \bar{a}_I}{s_i} \tag{5}$$

$$\bar{a}_I = \frac{1}{n} \sum_{j=0}^{n-1} a_{i,j} \tag{6}$$

$$s_i = \sqrt{\frac{1}{j-1} \sum_{j=0}^{n-1} a_{i,j} - \bar{a}_I} \tag{7}$$

Compared with other methods, our proposed data processing method is normalized according to the row dimension of matrix A, while the traditional data processing method is to process data according to the column dimension of matrix A. The morphological characteristics of the ECG cardiac signal are preserved as much as possible using our technology.

We chose NE data from the MIT-BIH database and the St. Petersburg database and plotted them to observe the impact of several pre-processing procedures. As shown in Fig 1, Fig 1(a) and 1(b) are the raw ECG data of the NE heartbeats of the two databases, respectively. As can be seen from the figure, the amplitudes of NE heartbeats in the two databases fluctuate greatly, and the fluctuations in the St-Petersburg databases are particularly pronounced. Such a directly merged database used to train neural networks will lead to problems such as difficult model convergence and poor training results. Fig 1(c) and 1(d) represent NE heartbeats data that have undergone traditional standardized processing, respectively. For traditional normalization, each sample subtracts the uniform global mean and divides it by the global standard deviation, and the actual effect is to move all ECG waveforms up and down as a whole and then scale them equally, so the amplitude fluctuation problem cannot be solved.

The heartbeat self-processing method can reduce the amplitude fluctuation and eliminate the influence of data statistical characteristic differences on the model recognition efficiency while maintaining the ECG morphological features. By its method, each heartbeat data subtracts its mean and is divided by its standard deviation. Compared to traditional standardization, this method is a local standardization method. Fig 1(e) and 1(f) represent locally standardized processing for NE-class heartbeats of MIT-BIH and St-Petersburg databases, respectively. The processed data is gathered together, which effectively solves the problem of amplitude fluctuation.

## 3.4 ECG databases integration methodology

On the basis of the study presented above, we suggest an ECG database integration approach in this section. The basic steps of this methodology are shown in Fig 2.

Step 1 clarifies the purpose of the ECG database. In this paper, we focus on arrhythmias detection and heartbeats classification.

Step 2 selects the ECG database according to the purpose. For heart rate detection and beat classification problems, there are many finely annotated ECG databases. In addition to the three selected ECG databases, we will also consider databases such as the sudden cardiac death Holter database [26], European ST-T database [27], and MIT-BIH ST change database [28] in the future.

Step 3 sets the heartbeats classes you focus on and the sampling number for each heartbeat, and selects the lead for each database.

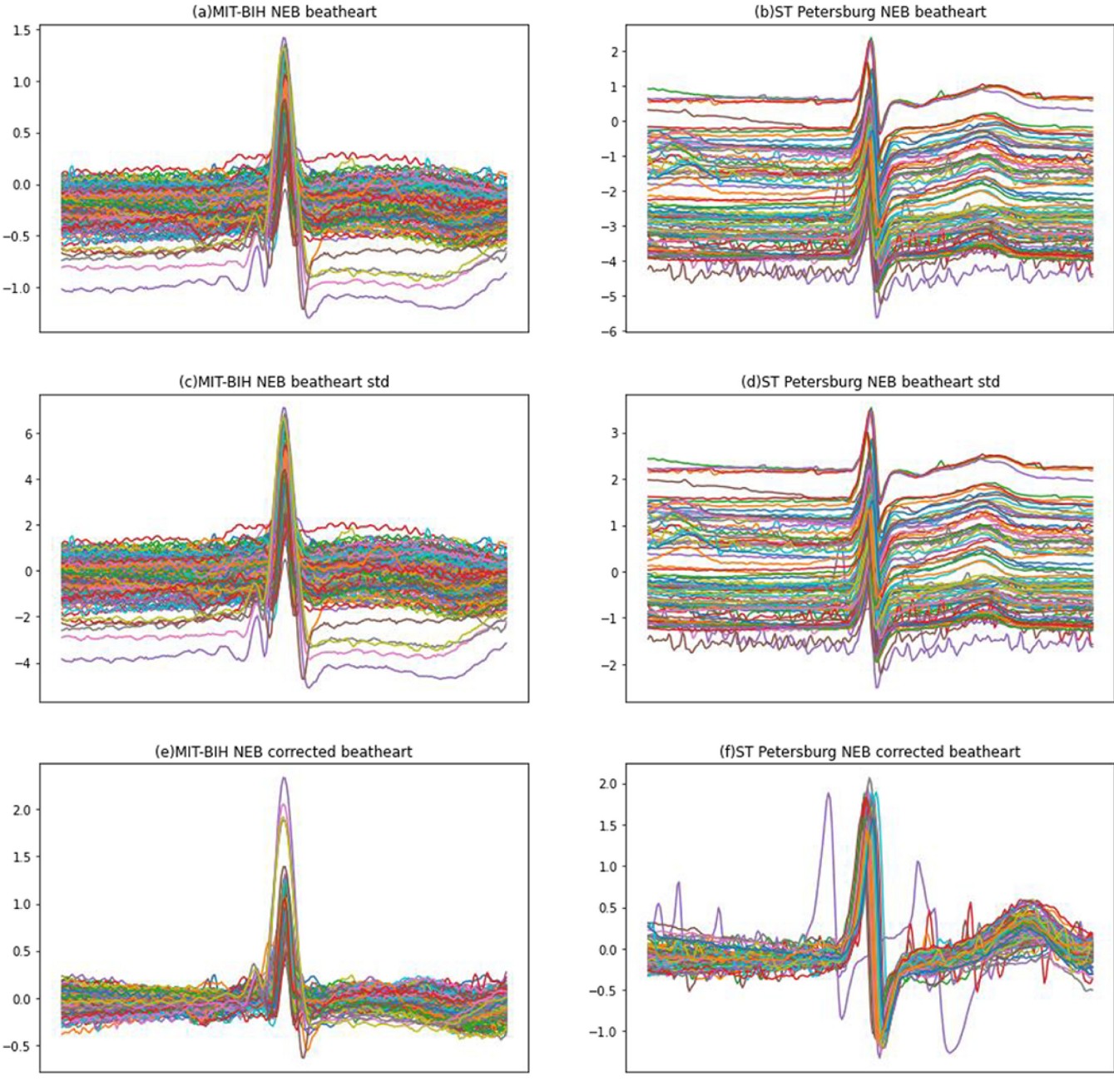

**Fig 1. Data per-processing.**

The heartbeats classes are used for targeted extraction of heartbeats, which determines which arrhythmia recognition training and testing use. The proper sampling number for each heartbeat facilitates complete and non-redundant extraction of heartbeats. Since different databases have data from different leads, choosing proper leads can make all the data from the same lead or similar leads as possible, which makes the data more consistent.

The number of heartbeats in normal people is 60 to 100 heartbeats per minute, of which more people have 70-80 heartbeats per minute. An adult's number of heartbeats exceeding 100

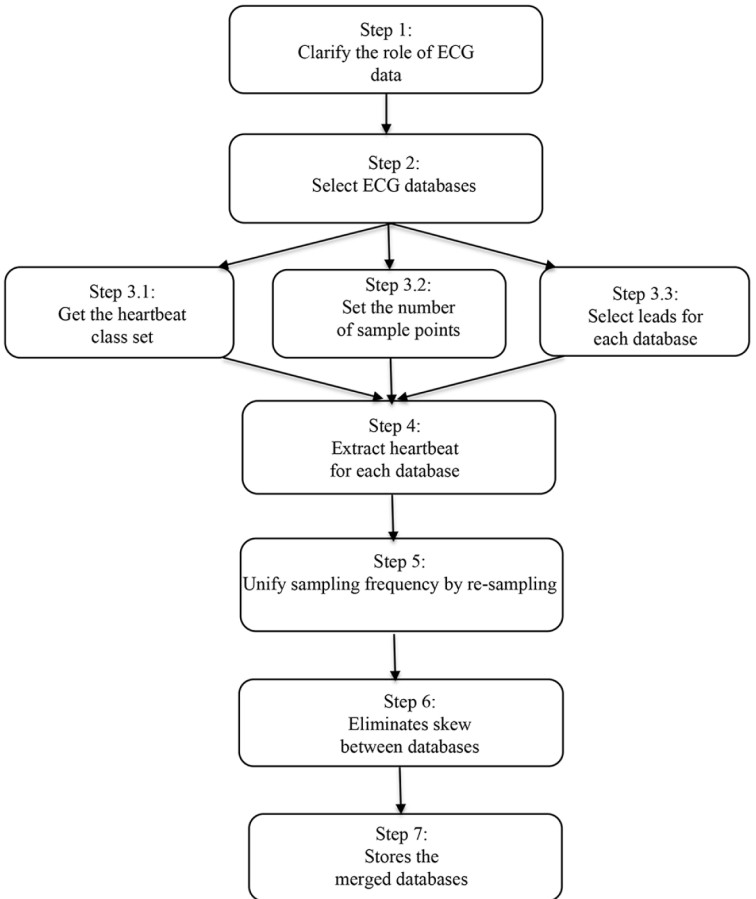

**Fig 2. ECG databases integration methodology.**

per minute is called tachycardia, and less than 60 beats per minute are called bradycardia. The frequency of the heartbeat is affected by factors such as age, gender, mood, and exercise. Usually, children's heart rates are faster than those of adults, and women's beat rates are faster than men's beat rates. The proper number of sampling points is calculated as follows.

Assuming that the heartbeat frequency is $f_{heart}$ and the sampling frequency is $f_{sample}$, the number of sampling points num_of_sample is calculated as follows:

$$num\_of\_sample = \frac{60}{f_{heart}} f_{sample} \tag{8}$$

$$60 \leq f_{heart} \leq 100 \tag{9}$$

$$f_{sample} \geq num\_of\_sample \geq 0.6 f_{sample} \tag{10}$$

Assuming a sampling frequency of 250, the number of sampling points per heartbeat should be 250 to 150 points. In this paper, we resample the other two databases at the MIT-BIH's sampling frequency. Since the sampling frequency is 360, we set the number of sampling points to a value between 360 and 216 (0.6*360), such as 260.

Step 4 is to extract heartbeats. A heartbeat consists of P, Q, R, S, and T peaks [4]. The specific method for extracting the heartbeat is to first read the position information of the R peak of the heartbeat from the ECG database, then read M sampling points in the past, and read N points backward. For a single heartbeat sampling point of 260, M is 129, and N is 130.

Step 5 is to resample the data so that all data have the same sampling frequency. Resampling is divided into up-sampling and down-sampling. The signal needs to be decimated when down-sampling, and the signal needs to be interpolated when up-sampling. Signal decimation is the process of decreasing the sample rate to eliminate extraneous data, whereas signal interpolation is the process of raising the sample rate to increase more data. Decimation, interpolation, and a combination of the two can be used to convert the sample rate of the signal. In this paper, we up-sample the other two databases to the sampling frequency of MIT-BIH.

Step 6 is data pre-processing to eliminate data bias. At this stage, a variety of pre-processing data methods, such as traditional labeling, extreme value, and self-healing processing methods, are available.

Step 7 saves training and testing databases. In the process of database segmentation, each class of heartbeat sample has a training and test set. We set the split ratio to 8 to 2, i.e., so 80% are utilized for training and 20% for testing.

## 4 Deep neural networks

To verify the validity of the integrated data set, we trained and tested classical neural networks. Fig 3 shows the first 5-layer neural network. It has an input layer, four fully connected hidden layers, and one output layer. The input layer's input units are equal to the number of sample points per heartbeat, such as 260. The hidden layer has 256 synapses and the dropout probability is 30%. The number of neurons in the output layer is 16.

The input data of the fully connected layer has been standardized in batches. All hidden layers use ReLU as their activation function. The SoftMax activation function is used in the last layer. Adam optimization and cross-entropy are used as cost functions.

We also build a 12-layer ResNet network. It contains 3 structurally identical residual blocks with a number of channels of 64, 128 and 128. Each residual block is composed of three 1-dimensional convolutional layers, and their convolution kernel sizes are 8×8, 5×5 and 3×3, respectively. The first two residual blocks are followed by a 1×1 convolutional layer for down

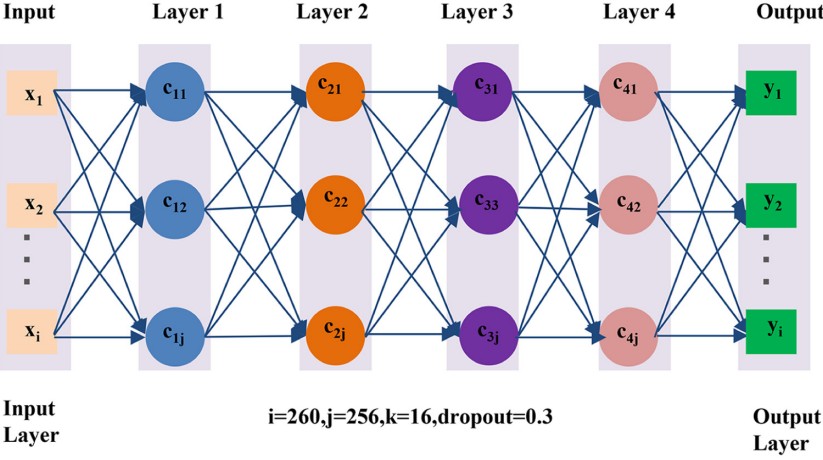

**Fig 3. The 5-layers fully connected neural network.**

sampling. All convolutional layers are equipped with batch normalization and ReLU activation layers. The fully connected classification layer and the Sofmax activation layer are at the end of the network.

## 5 Experiment

This work runs on a deep learning framework of Keras with Tensorflow as the backend. The workstation used consists of 6144 MB GPU (NVIDIA GeForce RTX-2060), Intel i7–6700K processor (4 GHz) and 16 GB RAM. Due to the limited computing power of the computer, we train for a maximum of 200 epochs and a learning rate of 0.001.

### 5.1 Test results

Table 4 details the test results for Hercules-3 in a 5-layer fully connected network, including 16 classes of precision, recall, F1-scores, and Num of Supported. From the test results, it can be seen that for 16 classification problems, the network achieved very high accuracy and recall rate.

Among them, the accuracy of 6 classes is as high as 100%, the accuracy of 1 class is 99%, and the accuracy of 2 classes is 98%. In addition to the high accuracy, the network's recall rate is also very high. For example, the recall rate for five classes is higher than 99%. To our knowledge, for the ECG Signal 16 classification network, this is currently the highest accuracy and recall.

It is worth mentioning that the sample distribution of the Hercules-3 database varies greatly. The number of samples of a single heartbeat class varies from single digits to tens of thousands, and the difference between different classes is six orders of magnitude. If you take measures such as sample expansion, you may further optimize the network performance. Since we mainly verify the validity of the synthesized database, the network performance is not fully optimized. These network test results fully prove that the integrated data set is valid and reasonable.

**Table 4. Test results of a 5-layer fully connected network.**

| Label | Precision | Recall | F1-score | Support |
|---|---|---|---|---|
| 0 | 0.99 | 1.00 | 0.99 | 77501 |
| 1 | 0.98 | 0.94 | 0.96 | 196 |
| 2 | 1.00 | 1.00 | 1.00 | 3 |
| 3 | 1.00 | 1.00 | 1.00 | 1405 |
| 4 | 0.89 | 0.66 | 0.76 | 64 |
| 5 | 1.00 | 0.17 | 0.29 | 6 |
| 6 | 1.00 | 0.99 | 1.00 | 1614 |
| 7 | 1.00 | 0.99 | 1.00 | 2086 |
| 8 | 0.89 | 0.81 | 0.85 | 2440 |
| 9 | 0.94 | 0.89 | 0.92 | 898 |
| 10 | 0.79 | 0.83 | 0.81 | 18 |
| 11 | 0.83 | 0.67 | 0.74 | 30 |
| 12 | 0.98 | 0.97 | 0.97 | 7414 |
| 13 | 1.00 | 0.90 | 0.95 | 21 |
| 14 | 0.84 | 0.72 | 0.78 | 209 |
| 15 | 0.50 | 0.08 | 0.14 | 24 |

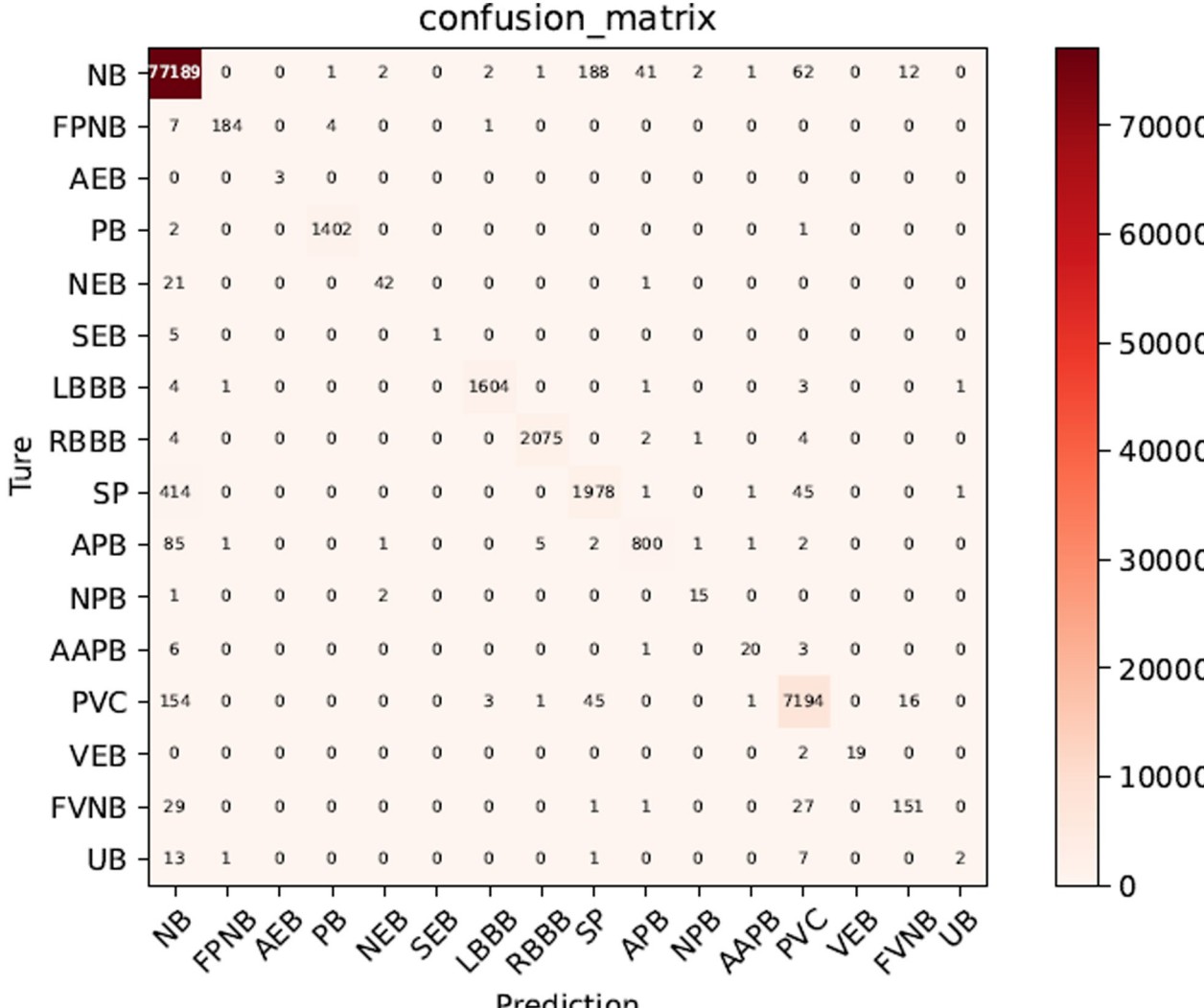

**Fig 4. Confusion matrix of our test.**

To further visualize the neural network performance, we list the confusion matrices of the test results, as shown in Fig 4. The specific recall rate can be calculated by the confusion matrix. For example, the recall rate of NB is actually 77189/77501, which is about 99.597%. The test database has a total of 93,929 samples, of which 92,679 samples are correctly predicted, with a prediction accuracy of 98.67%. For classes with small data, Out of 3 AEB samples, 100% are correctly classified as AEB class. Out of 30 AAPB ECG samples, 93.13% are correctly classified as AAPB class.

## 5.2 Data pre-processing

In order to verify the effect of the heartbeat self-processing method, we compared it to other data processing methods, and the results are shown in Table 5. It includes the performance indicators given by a 5-layer fully connected network under no pre-processing, Standardization, extremum, averaging, or standard deviation.

**Table 5. Data pre-processing.**

| Methods | misclassified samples | Accuracy (%) | Class Recall | Class Accuracy | F1-score |
|---|---|---|---|---|---|
| Self-processing | 1250 | 98.67 | 0.79 | 0.89 | 0.82 |
| NoPre-processing | 1485 | 98.41 | 0.69 | 0.84 | 0.75 |
| Standardization | 1551 | 98.34 | 0.72 | 0.82 | 0.76 |
| Extremum | 1493 | 98.41 | 0.71 | 0.82 | 0.76 |
| Averaging | 1381 | 98.52 | 0.70 | 0.82 | 0.75 |
| Standard deviation | 1369 | 98.54 | 0.70 | 0.85 | 0.76 |

From this table, it can be seen that our proposed method outperforms other pre-processing methods in all metrics. Our proposed method improves class recall by at least 6%, class accuracy by at least 4%, and F1-score by at least 7%.

## 5.3 Performance comparison

We offer the most recent findings from related studies combined with our experimental results in order to compare them to others' findings, as shown in Table 6. This table focuses on comparing the ECG databases used in these papers, the number of heartbeat classes, models, and classification accuracy. The most popular ECG database is the MIT-BIH database, as seen in the table. Many papers have studied fewer heartbeat classes.

References [20, 23, 24] are all the public papers I could find that classify heartbeats into 16 classes. It can be seen from the comparison that we adopt a relatively simple network structure for the largest database and achieve relatively ideal results.

## 5.4 Inter-patient test

To verify the performance of 5-layer fully connected network model, we used the Sudden cardiac death Holter database for an inter-patient experiment. This database is a collection of long-term ECG recordings of patients who experienced sudden cardiac death during the recordings. These recordings were mainly obtained in Boston hospitals. The database includes 23 patients with underlying sinus rhythm, continuously paced, or atrial fibrillation. All

**Table 6. Compare with other papers.**

| | ECG database | Number of classes | models | Accuracy |
|---|---|---|---|---|
| [22] | MIT-BIH | 3 | random forest | 88.7% |
| [29] | ChapmanECG [25] | 3 | ResNet | 99.98% |
| [30] | MIT-BIH | 4 | SVM/CNN | 99.21% |
| | cardiac arrhythmia database | | | 99.39% |
| [31] | MIT-BIH | 4 | CNN-LSTM+RRHOS-LSTM | 99.16% |
| [32] | MIT-BIH Atrial Fibrillation Database | 9 | CNN+LSTM+GRU | 99.01% |
| [33] | CPSC 2018 | 9 | CNN | 82.5% |
| [20] | MIT-BIH | 16 | SVM | 99.3% |
| [23] | MIT-BIH MIT-BIH-Sup | 16 | CNN | 92.73% |
| [24] | MIT-BIH | 16 | Sine–Cosine Algorithm | 99.2% |
| Our method | Hercules-3 | 16 | 9-layer CNN | 98.01% |
| Our method | Hercules-3 | 16 | 5-layer FC Network | 98.67% |

heartbeats of this database were used as test data, and the accuracy of the fully connected network structure test was 96.62%.

The 12-layer ResNet network was trained using the training dataset proposed in this paper, and the test dataset was used for intra-patient testing, and finally the Sudden cardiac death holter database was used for an inter-patient experiment. The 12-layer ResNet achieved an accuracy rate of 97.39% in intra-patient experiments and 96.64% accuracy in inter-patient experiments, and its performance is similar as the fully connected networks. Compared with ResNet, the fully connected network has a shorter training time and shorter testing time.

Furthermore, we used MIT-BIH and MIT-BIH-SUP as the training set and ST-Petersburg as the test set, and the 5-layer fully connected network model to verify the effectiveness of the proposed method. Under the same training parameters (the training period was 30, the cost function was cross-entropy, the batch size was 300, and the validation data accounted for 30% of the training set), We compared the five data processing methods mentioned in section 3.3. The test results show that, the test accuracy was 91.7% for our method, 88.76% for Standardization, 87.56% for Extreme value, 88.74% for Averaging, 87.82% for Standard deviation, 87.55% without any standardization. Our data processing method outperformed other methods. The effect of other methods was similar to the effect of no-standardizing because there are batch standardization layers within the network. However, our method eliminated variability across samples and databases, and was orthogonal to other standardized methods, so it worked better.

## 6 Conclusion

The most intuitive way to build a large ECG database is to integrate multiple ECG databases together. However, there are many problems with simply merging databases together, such as statistical differences in the data. To this end, in this paper we propose a multi-database integration methodology and a heartbeat self-processing method to eliminate data differences. We integrated the three databases together and noted Hercules-3. In order to verify its practicality, we train and test a 16-classification 5-layer fully connected neural network based on Hercules-3 and achieve an accuracy rate of up to 98.67%. Compared with other pre-processing methods, our proposed method improves class recall by at least 6%, class accuracy by at least 4%, and F1-score by at least 7%. The experimental results show that the integrated database is more effective than a single database.

## Author Contributions

**Conceptualization:** Meng Qi, Hongxiang Shao.

**Data curation:** Meng Qi, Nianfeng Shi.

**Formal analysis:** Guoqiang Wang.

**Funding acquisition:** Hongxiang Shao.

**Investigation:** Yifei Lv.

**Methodology:** Meng Qi.

**Resources:** Nianfeng Shi.

**Software:** Meng Qi, Nianfeng Shi, Guoqiang Wang.

**Supervision:** Meng Qi, Guoqiang Wang.

**Validation:** Meng Qi.

**Writing – original draft:** Meng Qi.

**Writing – review & editing:** Meng Qi, Hongxiang Shao.

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
