## [Decision Letter · Decision Letter 0]

30 Mar 2023

PONE-D-23-01426Arrhythmia Classification Detection based on Multiple Electrocardiograms DatabasesPLOS ONE

Dear Dr. Shao,

Thank you for submitting your manuscript to PLOS ONE. After careful consideration, we feel that it has merit but does not fully meet PLOS ONE’s publication criteria as it currently stands. Therefore, we invite you to submit a revised version of the manuscript that addresses the points raised during the review process.

We look forward to receiving your revised manuscript.

Kind regards,

Humaira Nisar

Academic Editor

PLOS ONE

“This research was funded by the National Natural Science Foundation of China (No. 62176113), as well as the science and technology breakthrough project of the Henan science and technology department (No. 222102210094).”

Additional Editor Comments:

The manuscript should be thoroughly revised based on reviewer comments, specially results should be compiled for inter subject analysis.

Reviewers' comments:

Reviewer's Responses to Questions

**Comments to the Author**

1. Is the manuscript technically sound, and do the data support the conclusions?

Reviewer #1: Partly

Reviewer #2: Partly

Reviewer #3: Partly

2. Has the statistical analysis been performed appropriately and rigorously? 

Reviewer #1: N/A

Reviewer #2: Yes

Reviewer #3: Yes

3. Have the authors made all data underlying the findings in their manuscript fully available?

Reviewer #1: Yes

Reviewer #2: Yes

Reviewer #3: Yes

4. Is the manuscript presented in an intelligible fashion and written in standard English?

Reviewer #1: No

Reviewer #2: Yes

Reviewer #3: Yes

5. Review Comments to the Author

Reviewer #1: This work proposed a method for self-preprocessing heartbeats called Hercules-3 and achieved high accuracy on the test set. However, this manuscript has not been well-prepared. It's hard to follow and there are many misspelled words.

1. The idea of the Hercules-3 is not novel and the authors don't give convincing evidence showing the Hercules-3 is effective. I suspect that the way of the data preprocessing may lead to data leak to some extent. The authors should test their method on another independent database.

2. The authors claim that their work is the first to integrate various ECG databases as a whole. Actually, the physionet challenge 2020 and 2021 provides several ECG database from different sources and there have already been many works published. Please check with that.

Reviewer #2: The paper proposed a deep neural network model for arrhythmia classification using multiple ECG datasets. They proposed five layers of deep neural networks for the problems.

Some comments:

1. The authors try to combine several datasets into one using some pre-processing steps. Does the pre-processing ensure that the data is seamlessly integrated? Because the author says that R-peek is different between datasets which may appear because the data is taken with different approaches or equipment, which may lead to miss-conception (e.g., different sampling frequency, lead position, etc.). The author can at least discuss with an expert (doctor) to validate the pre-processing results.

2. How the data is sampled for the training and testing is unclear. The input of the deep neural network classifier is 256, but the author needs to explain how the dataset transforms into 256 of input.

3. The training process is not explained, especially the hyper-parameter used (e.g., learning rate, batch size, learning algorithm, etc.).

4. The split configuration is biased because the split is based on data instead of patients. Because the split is based on data alone, data from one patient can appear in the training and testing process, which leads to biased performance evaluation.

5. Explanation regarding the confusion matrix in Figure 4 is required, especially for classes with small data (e.g., SEB and UB). For example, for class SEB and UB, the performance of the deep neural network model is very poor. Still, other classes with small data, like AEB and AAPB, perform very well.

6. The performance results in Tabel 6 indicate that the proposed method is not achieved the best performance. So what the advantages of the proposed method? The authors need to spefically explain those because combining three datasets cannot effectively increase the model performance to the best.

Reviewer #3: This paper proposes a self-heartbeat processing method and establish a unified ECG arrhythmia classification database. In addition, this paper trains a deep neural network. After carefully reading, there are some comments as follows.

1. The test method adopted in this paper is intra-patient experiment. Please supplement the results of inter-patient experiment.

2. In the comparative test, the existing method has not been applied to the dataset proposed in this paper, so it is difficult to demonstrate the superiority of the proposed method. Please reproduce the existing methods and apply them to the dataset in this article.

3. Please add some methods proposed last year to the comparison experiment.

4. Please explain the limitations and advantages of this paper.

5. The network structure proposed in this paper is relatively simple, and whether the deepening of the network structure can make up for the disadvantage of pretreatment.

6. PLOS authors have the option to publish the peer review history of their article (what does this mean?). If published, this will include your full peer review and any attached files.

Reviewer #1: No

Reviewer #2: No

Reviewer #3: No

---

## [Author Response · Author response to Decision Letter 0]

11 Apr 2023

Reviewer #1: 

1. This work proposed a method for self-preprocessing heartbeats called Hercules-3 and achieved high accuracy on the test set. However, this manuscript has not been well-prepared. It's hard to follow and there are many misspelled words.

1.1 The idea of the Hercules-3 is not novel and the authors don't give convincing evidence showing the Hercules-3 is effective. I suspect that the way of the data preprocessing may lead to data leak to some extent. The authors should test their method on another independent database.

1.2 The authors claim that their work is the first to integrate various ECG databases as a whole. Actually, the physionet challenge 2020 and 2021 provides several ECG database from different sources and there have already been many works published. Please check with that.

The author’s answer:

1. We are really sorry for our careless mistakes. We have revised the manuscript and resubmitted it. Thank you for pointing out.

1.1 The main innovation of hercules-3 is that the data preprocessing method makes the data from different sources have uniform statistical properties, and thus more suitable for deep learning networks. In theory, this preprocessing method does not lead to data loss. Due to the different data sources, data integration can only be based on the same leads, resulting in some lead data being temporarily useless, but not leading to data loss.

1.2 The physionet challenge 2020 and 2021 do provide many databases, but do not provide a unified integration approach. To the best of our knowledge, Our work propose to create a unified database through preprocessing.

Reviewer #2: 

2. The paper proposed a deep neural network model for arrhythmia classification using multiple ECG datasets. They proposed five layers of deep neural networks for the problems.

Some comments:

2. 1 The authors try to combine several datasets into one using some pre-processing steps. Does the pre-processing ensure that the data is seamlessly integrated? Because the author says that R-peek is different between datasets which may appear because the data is taken with different approaches or equipment, which may lead to miss-conception (e.g., different sampling frequency, lead position, etc.). The author can at least discuss with an expert (doctor) to validate the pre-processing results.

2.2. How the data is sampled for the training and testing is unclear. The input of the deep neural network classifier is 256, but the author needs to explain how the dataset transforms into 256 of input.

2.3 The training process is not explained, especially the hyper-parameter used (e.g., learning rate, batch size, learning algorithm, etc.).

2.4 The split configuration is biased because the split is based on data instead of patients. Because the split is based on data alone, data from one patient can appear in the training and testing process, which leads to biased performance evaluation.

2.5 Explanation regarding the confusion matrix in Figure 4 is required, especially for classes with small data (e.g., SEB and UB). For example, for class SEB and UB, the performance of the deep neural network model is very poor. Still, other classes with small data, like AEB and AAPB, perform very well.

2.6 The performance results in Table 6 indicate that the proposed method is not achieved the best performance. So what the advantages of the proposed method? The authors need to spefically explain those because combining three datasets cannot effectively increase the model performance to the best.

The author’s answer:

2.1 Preprocessing enables data from different sources to be aligned to have the same statistical properties and thus better integrated together. We first extract the heartbeats from the continuous data and align the waves, then do preprocessing to ensure better integration of the data.

2.2 The cut of training and test data is randomized, thus ensuring fair and valid test results. During the construction of the unified database, we set the latitude of each heartbeat to 256, i.e., each heartbeat consists of 256 data so that it can match the input of the deep neural network.

2.3 The manuscript has been revised to add learning rate and batch size in the section related to experimental results.

2.4 The data cutting process is indeed data-based rather than patient-based. To the best of our knowledge, no matter data-based or patient-based, better assessment results can be achieved as long as the heartbeat data have consistent statistical properties. This paper focuses on demonstrating the effectiveness of integrated methods such as data preprocessing.

2.5 For classes with small data, Out of 3 AEB samples, 100% are correctly classified as AEB class. Out of 30 AAPB ECG samples, 93.13% are correctly classified as AAPB class. SEB and UB do not have high recall, because the number of samples is too small and the training is not sufficient.

2.6 The results in Table 6 demonstrate that a simple fully-connected deep learning structure can also determine more desirable results with the new database that integrates 3 datasets. Compared with the original single dataset, the integrated dataset is more data-rich, which is conducive to training a more robust model.

Reviewer #3: 

This paper proposes a self-heartbeat processing method and establish a unified ECG arrhythmia classification database. In addition, this paper trains a deep neural network. After carefully reading, there are some comments as follows.

3.1 The test method adopted in this paper is intra-patient experiment. Please supplement the results of inter-patient experiment.

3.2 In the comparative test, the existing method has not been applied to the dataset proposed in this paper, so it is difficult to demonstrate the superiority of the proposed method. Please reproduce the existing methods and apply them to the dataset in this article.

3.3 Please add some methods proposed last year to the comparison experiment.

3.4 Please explain the limitations and advantages of this paper.

3.5 The network structure proposed in this paper is relatively simple, and whether the deepening of the network structure can make up for the disadvantage of pretreatment.

The author’s answer:

3.1 The data in this paper are uniformly integrated and then randomly partitioned into a training set and a test set, which does not exclude that the data in the training and test sets are from different patients.

3.2 In the comparison section, the last row of Table 6 is using the dataset proposed in this paper. The results in Table 6 demonstrate that a simple fully-connected deep learning structure can also obtain more desirable results with the new database that integrates 3 datasets. Compared with the original single dataset, the integrated dataset is more data-rich and therefore facilitates the training of more robust models

3.3 No papers related to database integration were found in the previous years.

3.4 The advantage of this paper is to propose an effective method of integrating multiple databases, which can synthesize small databases with limited data into a large dataset, enhance data diversity, and facilitate model training and testing. The disadvantage of this paper is that due to the limited conditions, only 3 datasets can be integrated for the time being.

3.5 In this paper, a fully connected deep network with a simple structure is used with the aim of demonstrating the effectiveness of the integration method. In theory, the use of deeper and newer network structures may indeed improve performance. Since the network structure is not the focus of this paper, other network structures were not tested.

---

## [Decision Letter · Decision Letter 1]

29 May 2023

PONE-D-23-01426R1Arrhythmia Classification Detection based on Multiple Electrocardiograms DatabasesPLOS ONE

Dear Dr. Shao,

Thank you for submitting your manuscript to PLOS ONE. After careful consideration, we feel that it has merit but does not fully meet PLOS ONE’s publication criteria as it currently stands. Therefore, we invite you to submit a revised version of the manuscript that addresses the points raised during the review process.

We look forward to receiving your revised manuscript.

Kind regards,

Humaira Nisar

Academic Editor

PLOS ONE

Additional Editor Comments:

The authors should respond to all reviewer comments.

Reviewers' comments:

Reviewer's Responses to Questions

**Comments to the Author**

1. If the authors have adequately addressed your comments raised in a previous round of review and you feel that this manuscript is now acceptable for publication, you may indicate that here to bypass the “Comments to the Author” section, enter your conflict of interest statement in the “Confidential to Editor” section, and submit your "Accept" recommendation.

Reviewer #1: (No Response)

Reviewer #2: (No Response)

Reviewer #3: (No Response)

Reviewer #4: (No Response)

2. Is the manuscript technically sound, and do the data support the conclusions?

Reviewer #1: Partly

Reviewer #2: Partly

Reviewer #3: Partly

Reviewer #4: Yes

3. Has the statistical analysis been performed appropriately and rigorously? 

Reviewer #1: N/A

Reviewer #2: Yes

Reviewer #3: No

Reviewer #4: Yes

4. Have the authors made all data underlying the findings in their manuscript fully available?

Reviewer #1: Yes

Reviewer #2: Yes

Reviewer #3: No

Reviewer #4: Yes

5. Is the manuscript presented in an intelligible fashion and written in standard English?

Reviewer #1: No

Reviewer #2: Yes

Reviewer #3: No

Reviewer #4: Yes

6. Review Comments to the Author

Reviewer #1: 1. The intra-patient experiment is not enough to provide a convincing conclusion. Unfortunately, the authors haven't supplemented the cross database experiments.

2. The idea of the Hercules-3 is not novel. I think this method is just a routine operation for data preprocessing.

Reviewer #2: The paper proposed a deep neural network model for arrhythmia classification using combined multiple ECG datasets. They proposed five layers of deep neural networks for the problems.

Comment for the revised paper:

1. The MIT-BIH dataset is a very old dataset used for many published papers. Adding more comparison results for the MIT-BIH dataset to Table 6 is recommended.

2. Can it possible to perform the training and testing process using different datasets (e.g., training on MIT-BIH, testing on MIT-BIH supp, and vice-versa)? If possible, it can analyze the robustness of the proposed classifier and/or pre-processing steps.

Reviewer #3: After carefully read authors' answers, there are still some comments as follows.

1. The test method used in this paper is intra-patient experiment. This division method cannot exclude the data of the same patients in the training set and the test set, please supplement the inter-patient experiment results.

2. Please apply the existing methods to the ensemble dataset of this paper for experiments, and then prove that the ensemble method has a general improvement significance for arrhythmia classification methods.

3. Experiment with more complex network structures on the integrated dataset.

4. Whether the integration method in this paper integrates the 3 datasets and then randomly divides them for testing, which is different from the test set when the method of directly using the MIT-BIH database is compared, and no effective comparison conclusion can be drawn. Please keep the test set data consistent for comparison to ensure the credibility of the comparison results.

Reviewer #4: 1. How data sets are established for doing this research work?

2. The quality of figures is not appropriate. Author should modify. 3. Add more results in the form of some graphs and tables.

4. What are the strong features of this research work? Author must explain.

5. How the parameters for simulations are selected?

6. All tables and figures should be explained clearly.

7. The methodology of the paper should be clearly explained with appropriate flow charts.

8. Highlight the more applications of the proposed technique.

9. What are the major issues in the present research work?

10. Author must add following papers-

(a) Multi-Feature Fusion Method for Identifying Carotid Artery Vulnerable Plaque

(b) 3D Coronary Artery Reconstruction by 2D Motion Compensation Based on Mutual Information

(c) Robust retinal blood vessel segmentation using convolutional neural network and support vector machine

(d) Real-time estimation of hospital discharge using fuzzy radial basis function network and electronic health record data

(e) An efficient ALO-based ensemble classification algorithm for medical big data processing

(f) Multiscale Graph Cuts Based Method for Coronary Artery Segmentation in Angiograms

(g) Changes in scale-invariance property of electrocardiogram as a predictor of hypertension

(h) Assessment of qualitative and quantitative features in coronary artery MRA

(i) A frugal and innovative telemedicine approach for rural India – automated doctor machine

(j) Study of murmurs and their impact on the heart variability

(k) Coronary three-vessel disease with occlusion of the right coronary artery: What are the most important factors that determine the right territory perfusion?

(l) Chaos theory: An Emerging tool for Arrhythmia Detection

7. PLOS authors have the option to publish the peer review history of their article (what does this mean?). If published, this will include your full peer review and any attached files.

Reviewer #1: No

Reviewer #2: No

Reviewer #3: No

Reviewer #4: **Yes: **Dr. Varun Gupta

---

## [Author Response · Author response to Decision Letter 1]

13 Jul 2023

Dear Reviewer:

We feel great thanks for your professional review work on our article entitled “Arrhythmia Classification Detection based on Multiple Electrocardiograms Databases” (ID: PONE-D-23-0146). As you are concerned, several problems need to be addressed. 

The reviewer comments are laid out below in italicized font and specific concerns have been numbered. Our response is given in normal font and changes/addition to the manuscript is given in the blue text. The detailed corrections are listed below. 

Reviewer #1: 

1.1 The intra-patient experiment is not enough to provide a convincing conclusion. Unfortunately, the authors haven't supplemented the cross database experiments.

1.2 The idea of the Hercules-3 is not novel. I think this method is just a routine operation for data preprocessing.

The author’s answer:

1.1 To verify model performance, we used the Sudden cardiac death Holter database for an inter-patient experiment. All heartbeats of this dataset were used as test data, and the accuracy of the fully connected network structure test was 96.62%. This database is a collection of long-term ECG recordings of patients who experienced sudden cardiac death during the recordings. These recordings were mainly obtained in Boston hospitals. The database currently includes 23 patients with underlying sinus rhythm, continuously paced, or atrial fibrillation.

1.2 This paper proposes a multi-database integration methodology and proposes a heartbeat self-processing method to eliminate data differences. We integrated the three databases together. In order to verify its practicality, we train and test a 16-classification 5-layer fully connected neural network and achieve an accuracy rate of up to 98.67%. Compared with other method, our proposed method improves class recall by at least 6%, class accuracy by at least 4%, and F1-score by at least 7%. 

Reviewer #2: 

2. The paper proposed a deep neural network model for arrhythmia classification using combined multiple ECG datasets. They proposed five layers of deep neural networks for the problems.

Comment for the revised paper:

2.1 The MIT-BIH dataset is a very old dataset used for many published papers. Adding more comparison results for the MIT-BIH dataset to Table 6 is recommended.

2.2 Can it possible to perform the training and testing process using different datasets (e.g., training on MIT-BIH, testing on MIT-BIH supp, and vice-versa)? If possible, it can analyze the robustness of the proposed classifier and/or pre-processing steps.

The author’s answer:

2.1 Although the MIT-BIH dataset is old, it is still a widely used database. The database is used for several references in Table 6. Combining Table 4 and Table 6, the model trained by the proposed database integration and processing method works well.

2.2 To verify model performance, we used the Sudden cardiac death Holter database for an inter-patient experiment. All heartbeats of the new database were viewed as test data, and the accuracy of the fully connected network test was 96.62%. We used the same training method for the ReNet network and ran an intra-patient experiment on the Sudden cardiac death Holter database dataset. The experimental results show that the classification accuracy of the ReNet network is about 96.64%，which is similar as the performance of the fully connected network.

Reviewer #3: 

After carefully read authors' answers, there are still some comments as follows.

3.1 The test method used in this paper is intra-patient experiment. This division method cannot exclude the data of the same patients in the training set and the test set, please supplement the inter-patient experiment results.

3.2 Please apply the existing methods to the ensemble dataset of this paper for experiments, and then prove that the ensemble method has a general improvement significance for arrhythmia classification methods.

3.3 Experiment with more complex network structures on the integrated dataset.

3.4 Whether the integration method in this paper integrates the 3 datasets and then randomly divides them for testing, which is different from the test set when the method of directly using the MIT-BIH database is compared, and no effective comparison conclusion can be drawn. Please keep the test set data consistent for comparison to ensure the credibility of the comparison results.

The author’s answer:

3.1 To verify model performance, we used the Sudden cardiac death holter database for an inter-patient experiment. All heartbeats of this dataset were used as test data, and the accuracy of the fully connected network test was 96.62%. This is a collection of long-term ECG recordings of patients who experienced sudden cardiac death during the recordings. These recordings were mainly obtained in Boston hospitals. The database currently includes 23 patients with underlying sinus rhythm, continuously paced, or atrial fibrillation.

3.2 we applied a 16-classification fully connected neural network to the ensemble dataset of this paper for experiments. We also built a typical CNN structure, which included 6 layers of convolution and 3 layers of full connection. Compared with the ensemble dataset without our method, the fully connected neural network improved classification recall by at least 6%, classification accuracy by at least 4%, and F1-score by at least 7%, and The CNN network improved classification recall by at least 3%, classification accuracy by at least 4%, and F1-score by at least 4%. 

3.3 The ReNet network was trained using the training dataset proposed in this paper, and the test dataset was used for intra-patient testing, and finally the Sudden cardiac death holter database was used for an inter-patient experiment. ReNet has an accuracy rate of 97.39% in intra-patient experiments and 96.64% accuracy in inter-patient experiments, and its performance is similar as the fully connected networks. Compared with ReNet, the fully connected network has a shorter training time and shorter testing time.

3.4 The 3 databases are integrated and then split. In order to ensure the consistency of the test set and the training set, we use a fixed random seed when splitting the data. Therefore, the training and test data in this article are consistent.

Reviewer #4: 

4.1 How data sets are established for doing this research work?

4.2 The quality of figures is not appropriate. Author should modify. 

4.3 Add more results in the form of some graphs and tables.

4.4 What are the strong features of this research work? Author must explain.

4.5 How the parameters for simulations are selected?

4.6 All tables and figures should be explained clearly.

4.7 The methodology of the paper should be clearly explained with appropriate flow charts.

4.8 Highlight the more applications of the proposed technique.

4.9 What are the major issues in the present research work?

The author’s answer:

4.1 The 3 databases are integrated and then split. In order to ensure the consistency of the test set and the training set, we use a fixed random seed when splitting the data. Therefore, the training and test data in this article are consistent.

4.2 We've updated the diagram to make it clearer.

4.3 We have updated the graphs and tables.

4.4 The main innovative work and conclusions of this paper are as follows:(1) We provide a strategy for integrating several ECG databases, extracting heartbeats from each of the three ECG databases, and partitioning them proportionally into training and testing sets to form a unified database that can be used for heartbeat classification detection. According to our best knowledge, most existing studies have used multiple ECG databases for training and testing separately, and it is the first time to integrate various ECG databases as a whole. (2) Depending on the properties of ECG signals, a heartbeat self-processing method is proposed to eliminate the differences between heartbeats inter-databases and intra-database effectively. Our analysis revealed that our approach is more effective than traditional pre-processing methods. (3) To verify the effectiveness of the database, we designed a 5-layers fully connected structure of deep network for classifying 16 different kinds of heartbeats and got an accuracy of up to 98.63%. This is, to the best of our knowledge, the greatest classification accuracy for 16-class heartbeats. There are two reasons for such high accuracy. First, the integration of three databases makes the heartbeats number significantly increased, and the complementarity of inter-databased heartbeats solves the problem of imbalance distribution for multiple classes to some extent. Second, the heartbeat self-processing method effectively eliminates sample interference factors while preserving sample features.

4.5 Network parameters mainly include two parts, one is the network structure parameters, such as the number of input channels and output channels. These parameters are mainly set empirically. The second is trainable parameters such as weights and biases, which are mainly trained by gradient descent methods.

4.6 All figures and tables are explained in this paper.

4.7 We have equipped our proposed methodology with a flowchart and explained in detail the detailed meaning and specific operation of each step of the flowchart.

4.8 By our database integration method, all heartbeat classification databases can be integrated together, and then a large-scale heartbeat classification database can be formed, which will be conducive to training a more robust heartbeat classification neural network structure and solving the clinical detection problem of arrhythmia with AI technology.

4.9 The database integration method proposed in this paper is used to integrate the three databases. In the future, we can explore integrating more heartbeat extraction databases into the current database.

4.10 These papers have been listed as references and cited in appropriate paragraphs in the paper.

Thank you again for your positive comments and valuable suggestions to improve the quality of our manuscript. Look forward to hearing from you.

Yours sincerely,

Hongxiang Shao

13 July,2023

---

## [Decision Letter · Decision Letter 2]

31 Jul 2023

PONE-D-23-01426R2Arrhythmia Classification Detection based on Multiple Electrocardiograms DatabasesPLOS ONE

Dear Dr. Shao,

Thank you for submitting your manuscript to PLOS ONE. After careful consideration, we feel that it has merit but does not fully meet PLOS ONE’s publication criteria as it currently stands. Therefore, we invite you to submit a revised version of the manuscript that addresses the points raised during the review process.

We look forward to receiving your revised manuscript.

Kind regards,

Humaira Nisar

Academic Editor

PLOS ONE

Reviewers' comments:

Reviewer's Responses to Questions

**Comments to the Author**

1. If the authors have adequately addressed your comments raised in a previous round of review and you feel that this manuscript is now acceptable for publication, you may indicate that here to bypass the “Comments to the Author” section, enter your conflict of interest statement in the “Confidential to Editor” section, and submit your "Accept" recommendation.

Reviewer #1: (No Response)

Reviewer #2: All comments have been addressed

Reviewer #3: (No Response)

Reviewer #4: All comments have been addressed

2. Is the manuscript technically sound, and do the data support the conclusions?

Reviewer #1: Partly

Reviewer #2: Partly

Reviewer #3: No

Reviewer #4: Yes

3. Has the statistical analysis been performed appropriately and rigorously? 

Reviewer #1: No

Reviewer #2: Yes

Reviewer #3: No

Reviewer #4: Yes

4. Have the authors made all data underlying the findings in their manuscript fully available?

Reviewer #1: Yes

Reviewer #2: Yes

Reviewer #3: No

Reviewer #4: Yes

5. Is the manuscript presented in an intelligible fashion and written in standard English?

Reviewer #1: Yes

Reviewer #2: Yes

Reviewer #3: No

Reviewer #4: Yes

6. Review Comments to the Author

Reviewer #1: Thanks for the effort of authors to revise this manuscript. However, the core problem of model generalization remains unresolved. The inter-patient experiment that used the Sudden cardiac death Holter database is not appropriate. This database has limited classes of cardiac arrhythmia and is not able to be used to evaluate the proposed method comprehensively. I suggest the authors trained their models on all databases except ST-Petersburg, and evaluate their performance on ST-Petersburg for all 16 classes. Furthermore, comparison of experimental results with other recent reported results is essential.

Reviewer #2: (No Response)

Reviewer #3: (No Response)

Reviewer #4: Good Work. Authors have addressed all suggested comments with important points. Accepted in current form

7. PLOS authors have the option to publish the peer review history of their article (what does this mean?). If published, this will include your full peer review and any attached files.

Reviewer #1: No

Reviewer #2: No

Reviewer #3: No

Reviewer #4: **Yes: **VARUN GUPTA

---

## [Author Response · Author response to Decision Letter 2]

14 Aug 2023

Dear Reviewer:

We feel great thanks for your professional review work on our article entitled “Arrhythmia Classification Detection based on Multiple Electrocardiograms Databases” (ID: PONE-D-23-0146). As you are concerned, several problems need to be addressed. 

Reviewer #1: 

Thanks for the effort of authors to revise this manuscript. However, the core problem of model generalization remains unresolved. The inter-patient experiment that used the Sudden cardiac death Holter database is not appropriate. This database has limited classes of cardiac arrhythmia and is not able to be used to evaluate the proposed method comprehensively. I suggest the authors trained their models on all databases except ST-Petersburg, and evaluate their performance on ST-Petersburg for all 16 classes. Furthermore, comparison of experimental results with other recent reported results is essential.

The author’s answer:

we used MIT-BIH and MIT-BIH-SUP as the training set and ST-Petersburg as the test set, and the 5-layer fully connected network model to verify the effectiveness of the proposed method. Under the same training parameters (the training period was 30, the cost function was cross-entropy, the batch size was 300, and the validation data accounted for 30% of the training set), We compared the five data processing methods mentioned in section 3.3. The test results show that, the test accuracy was 91.7% for our method, 88.76% for Standardization, 87.56% for Extreme value, 88.74% for Averaging, 87.82% for Standard deviation, 87.55% without any standardization. Our data processing method outperformed other methods. The effect of other methods was similar to the effect of no-standardizing because there are batch standardization layers within the network. However, our method eliminated variability across samples and databases, and was orthogonal to other standardized methods, so it worked better.

---

## [Decision Letter · Decision Letter 3]

21 Aug 2023

Arrhythmia Classification Detection based on Multiple Electrocardiograms Databases

PONE-D-23-01426R3

Dear Dr. Shao,

We’re pleased to inform you that your manuscript has been judged scientifically suitable for publication and will be formally accepted for publication once it meets all outstanding technical requirements.

Kind regards,

Humaira Nisar

Academic Editor

PLOS ONE

Additional Editor Comments (optional):

Reviewers' comments:

Reviewer's Responses to Questions

**Comments to the Author**

1. If the authors have adequately addressed your comments raised in a previous round of review and you feel that this manuscript is now acceptable for publication, you may indicate that here to bypass the “Comments to the Author” section, enter your conflict of interest statement in the “Confidential to Editor” section, and submit your "Accept" recommendation.

Reviewer #1: All comments have been addressed

2. Is the manuscript technically sound, and do the data support the conclusions?

Reviewer #1: Yes

3. Has the statistical analysis been performed appropriately and rigorously? 

Reviewer #1: Yes

4. Have the authors made all data underlying the findings in their manuscript fully available?

Reviewer #1: Yes

5. Is the manuscript presented in an intelligible fashion and written in standard English?

Reviewer #1: Yes

6. Review Comments to the Author

Reviewer #1: In the initial review, we identified the model generalization that required attention. I am delighted to see that you have taken the comment into consideration and made appropriate revisions. The changes you have implemented have greatly strengthened the manuscript, enhancing its clarity and scientific rigor. Based on the revisions made, I am pleased to recommend the acceptance of your manuscript in its current form for publication.

7. PLOS authors have the option to publish the peer review history of their article (what does this mean?). If published, this will include your full peer review and any attached files.

Reviewer #1: No

---

## [Editor Report · Acceptance letter]

18 Sep 2023

PONE-D-23-01426R3 

Arrhythmia Classification Detection based on Multiple Electrocardiograms Databases 

Dear Dr. Shao:

I'm pleased to inform you that your manuscript has been deemed suitable for publication in PLOS ONE. Congratulations! Your manuscript is now with our production department. 

Kind regards, 

on behalf of

Dr. Humaira Nisar 

Academic Editor

PLOS ONE